# A Practical Insecticide Resistance Monitoring Bioassay for Orally Ingested Dinotefuran in Anopheles Malaria Vectors

**DOI:** 10.3390/insects13040311

**Published:** 2022-03-22

**Authors:** George John Ian Parsons, Rosemary Susan Lees, Sofia Balaska, John Vontas

**Affiliations:** 1Vector Biology Department, Liverpool School of Tropical Medicine, Pembroke Place, Liverpool L3 5QA, UK; george.parsons@lstmed.ac.uk; 2Innovation to Impact, Pembroke Place, Liverpool L3 5QA, UK; 3Institute of Molecular Biology and Biotechnology, Foundation for Research and Technology-Hellas, 73100 Heraklion, Greece; blsophia94@gmail.com (S.B.); vontas@imbb.forth.gr (J.V.); 4Department of Biology, University of Crete, Vassilika Vouton, 71409 Heraklion, Greece; 5Pesticide Science Laboratory, Department of Crop Science, Agricultural University of Athens, 11855 Athens, Greece

**Keywords:** insecticide resistance, Attractive Toxic Sugar Bait (ATSB), Attractive Targeted Sugar Bait (ATSB), diagnostic bioassay, resistance monitoring

## Abstract

**Simple Summary:**

When insecticides are used to control mosquitoes, resistance is likely to develop over time. It is important to monitor the trait so that an alternative insecticide class can be deployed if needed, to sustain the efficiency of the intervention. Most insecticides for control of adult malaria vectors are used in treated bed nets or sprayed on walls where mosquitoes rest, so that mosquitoes contact them through their tarsi (feet). To control mosquitoes which are becoming resistant to these tools, new insecticide-based tools using both different chemistry and mode of uptake have been developed. One example of these is Attractive Toxic Sugar Baits (ATSBs), from which mosquitoes feed and ingest insecticide that kills them. However, different methods may be needed to monitor for resistance against interventions that have different modes of uptake. This study employed a method for applying insecticide directly onto a mosquito and measuring mortality, and the results were related to mortality from the same insecticide when ingested. This demonstrated that the method may be suitable to detect signs of resistance developing in mosquito populations targeted with ATSBs. Application of the method in wild populations will provide further validation.

**Abstract:**

Attractive Toxic Sugar Baits (ATSB) deployed outdoors are likely to be particularly effective against outdoor biting mosquitoes and, if they contain insecticides with a different mode of action, mosquitoes resistant to pyrethroids. One such ATSB based on the neonicotinoid dinotefuran is currently under evaluation in Africa. As with any insecticide-based intervention, it will be important to monitor for the possible emergence of vector resistance. While methods for detecting resistance to insecticides via tarsal contact are recommended by the World Health Organization (WHO), these may not be applicable for orally ingested insecticides. Here, a new ingestion assay, appropriate for a controlled laboratory setting, is described using fluorescein sodium salt (uranine) as a feeding marker. Conventional topical application bioassays, more appropriate for routine deployment, have also been used to apply dinotefuran to the thorax of adult *Anopheles* mosquitoes with an organic carrier to bypass lipid cuticle barriers. The two methods were compared by establishing lethal doses (LD) in several *Anopheles* strains. The similarity of the ratios of susceptibility to dinotefuran between pairs of pyrethroid susceptible and resistant strains validates topical application as a suitable, more practical and field applicable method for monitoring for the emergence of resistance to orally ingested dinotefuran. A discriminating dose is proposed, which will be further validated against field populations and used to routinely monitor for the emergence of resistance alongside ATSB trials.

## 1. Introduction

The prevention of vector borne diseases is often achieved by controlling the insect population, which currently largely relies on the use of insecticides. Malaria prevalence has halved since 2000, primarily due to vector control interventions, saving 660 million lives, with a large part of the reduction being attributable to the use of insecticides [1]. The primary vector control tools employed against malaria are insecticide treated nets (ITNs) and indoor residual spray (IRS). However, insecticide resistance represents a major threat to human health. Alternative interventions with different active ingredients and/or modes of action, capable of controlling insecticide resistant vectors, as well as vectors which transmit malaria outdoors, are urgently required to ensure the sustainability of malaria control interventions.

A number of Attractive Toxic Sugar Baits (ATSBs) are being evaluated as part of an Integrated Vector Management (IVM) approach. ATSBs deployed outdoors are likely to be particularly effective against outdoor biting mosquitoes, as well as mosquitoes that are resistant to pyrethroid insecticides. An ATSB has been developed by Westham Co. which utilizes the neonicotinoid dinotefuran. The bait station includes a permeable membrane that allows volatile attractive compounds to be emitted and encourage feeding by mosquitoes yet minimizes feeding by non-target organisms (NTOs) and tarsal contact of both mosquitoes and NTOs with the insecticide-treated bait. Because of the inclusion of this permeable membrane, the product is termed an Attractive *Targeted* Sugar Bait (ATSB^®^) [2]. These bait stations have been shown to be effective in controlling malaria vectors in Mali [3] and are under evaluation in conjunction with the Innovative Vector Control Consortium (IVCC) in trials in Zambia, Kenya, and Mali. Dinotefuran is included as the active ingredient and, since this insecticide is new to public health, it is not expected that target mosquito populations will carry any resistance to it, though another neonicotinoid, clothianidin, is now in use for public health and cross-resistance is a risk. As with any new vector control tool based on insecticides, once dinotefuran-based ATSBs are deployed, susceptibility testing will need to be introduced to allow early detection of possible emerging resistance and enable evidence-based resistance management strategies. Conventionally discriminating (or diagnostic) dose bioassays are used to detect resistance to insecticides encountered by mosquitoes through tarsal contact on an ITN or IRS, and so a WHO tube assay [4] or the Centers for Disease Control and Prevention (CDC) bottle bioassay [5] is used. Survival in a discriminating dose (DD) or discriminating concentration (DC) assay is a sign of possible resistance in the target population.

Such diagnostic bioassays are not available for orally ingested insecticides, and nor is a DC recommended by the WHO for susceptibility monitoring of dinotefuran [4] via tarsal contact. Methods for detecting resistance to neurotoxic insecticides via tarsal contact may not be applicable for the orally ingested dinotefuran due to its negative log P, which inhibits tarsal uptake due to epicuticular lipids and barriers. In addition, it is possible that this different method of exposure may be affected by different resistance mechanisms than those responsible for resistance against contact insecticides. It is therefore desirable to establish a suitable method to screen for dinotefuran resistance in ATSB deployment sites.

Ideally, mosquitoes would be fed a discriminating dose of dinotefuran ingested in a sugar solution to most closely match the exposure route in an ATSB. However, to date only a few methods, rather complicated in terms of practical implementation, have been established to feed a spiked sugar meal to mosquitoes with a high enough feeding rate to allow this form of resistance monitoring to be done [6,7]. Assays based on feeding an AI to insects in a sugar meal are vulnerable to huge variability and poor accuracy, due to poor feeding rates, especially with recently colonized or field-caught mosquitoes, and variable volumes taken up by those mosquitoes that do feed. Topical application of insecticide bypasses tarsal barriers by applying insecticide solutions in lipophilic solvent directly to the thorax of the mosquito to be taken in through the cuticle [4]. Although this is not the same entry system as the proposed delivery via ingestion, it is a technique that also bypasses cuticular barriers and therefore may be sufficiently representative of oral uptake whilst being easily applicable for routine susceptibility monitoring in field sites. Indeed, it has been shown in agricultural pests that the response to exposure to neurotoxic insecticides by topical application is a good proxy for the response to ingestion and that resistance monitoring assays for oral insecticides can be based on topical application [8]. Topical application is relatively quick with even large numbers of mosquitoes and can be done with fairly straightforward portable equipment, and as such is a more robust method for susceptibility testing.

Here, an oral application assay has been developed, able to determine dose response curves among two *Anopheles* strains, that prevent tarsal contact while allowing ingestion of a spiked sugar meal. Practical topical application bioassays were also developed by applying dinotefuran to the thorax of adult *Anopheles* mosquitoes with organic carrier to bypass lipid cuticle barriers. The topical application dose response was compared with the oral toxicity bioassays across several *Anopheles* strains.

## 2. Materials and Methods

### 2.1. Mosquito Rearing

Mosquitoes were reared within the insectaries of Liverpool Insect Testing Establishment (LITE) at the Liverpool School of Tropical Medicine as previously described [9], at 26 ± 2 °C and 80 ± 10% relative humidity. Four strains were used for experiments, all described by Williams et al. [9]. Kisumu is an insecticide-susceptible strain of *Anopheles gambiae*, colonized from Kenya in 1975. VK7 2014 is a strain of *An. coluzzi* colonized from Valley de Kou, Burkina Faso, in 2014 and resistant to pyrethroids and Dichlorodiphenyltrichloroethane (DDT) as a result of both target site and metabolic resistance mechanisms. Fang is a susceptible colony of *An. funestus* colonized from Calueque, Southern Angola in 2015. FUMOZ-R, also *An. funestus,* was colonized from Mozambique in 2000 before being selected by exposure to lambda-cyhalothrin to produce a strain with a high level of metabolic resistance [10]. Though not of the same species, the VK7 2014 strain was compared to Kisumu, a model laboratory colony of the *Anopheles gambiae* species complex which is susceptible to all classes of insecticide. A direct species comparison of susceptible (Fang) and pyrethroid resistant (FUMOZ-R) strains was possible for *An. Funestus*.

All tests were carried out using 2–5 day old female mosquitoes which had been allowed to sugar feed and mate but not blood feed prior to testing. For mosquito size for each sample, refer to Appendix A.

### 2.2. Ingestion Assay

Between one hundred and two hundred 2–5 day old female mosquitoes were starved in a standard (30 cm × 30 cm × 30 cm) BugDorm-1 (MegaView Science Co., Ltd., Taichung, Taiwan) cage for approximately 18 h with ad libitum access to purified (Merck Millipore, Darmstadt, Germany) water-soaked cotton on top of the cage mesh. Mosquitoes were then exposed to insecticide mixed into sugar solution by adding 2 feeders, designed to prevent tarsal contact yet allow easy feeding, to each cage for 24 h (Figure 1).

The sugar solution was 10% sucrose (granulated sugar in de-ionized water), 0.8% Uranine (Fluorescein Sodium Salt; Honeywell, Charlotte, NC, USA) fluorescent marker, and treatment-dependent insecticide concentration (0.000001%, 0.00001%, 0.00002%, 0.000025%, 0.00005%, 0.000075%, 0.0001%, and 0.001% *w*/*v*). Between 1 and 5 replications were performed per concentration and between 5 and 9 replications for each control to collect sufficient data to generate LD values using Rstudio (See Appendix A). Technical grade (98.7%) dinotefuran was sourced from Sigma–Aldrich (Manchester, UK). 10 mL of each insecticide concentration solution was used to soak cotton wool inside sugar feeders less than 5 min prior to adding the feeder to the cage. Two feeders were used per cage to ensure mosquitoes had adequate access to sugar/insecticide solution.

After the exposure period, all mosquitoes were aspirated out of each cage into holding cups, separated by treatment as well as living or dead, then frozen at −20 °C. Once mosquitoes were killed (usually 1–2 h at freezing temperatures), they were scored for fluorescence using a Leica MZ 10 F microscope (Leica Microsystems, Milton Keynes, UK) under a yellow-fluorescent protein (YFP) filter (Figure 2). Only those mosquitoes that were scored as being positive for feeding by fluorescence were included in the mortality calculations. Feeding rate was calculated for each replicate test from the proportion of fluorescent positive mosquitoes relative to all exposed mosquitoes. All raw bioassay data is available in Appendix A.

### 2.3. Topical Application

Cohorts of 10 2–5 day old female mosquitoes at a time were knocked down using CO_2_ for 20 s before being transferred to a petri dish with filter paper dampened with purified water. While knocked down, the mosquitoes were positioned ventrally so that their dorsi were easily accessible. 0.2 μL aliquots of insecticide in acetone solution were applied to the dorsal side of each mosquito thorax using a 10 μL Hamilton syringe (Scientific Laboratory Supplies Ltd. (SLS), Nottingham, UK). Mosquitoes were then transferred back into holding cups and knock down or mortality was scored at 30 min, 60 min, and 24 h post-exposure. As well as an acetone-only negative control and a positive control of Permethrin at a concentration of 0.1%, six doses of dinotefuran were applied to Kisumu (0.0002%, 0.0005%, 0.0001%, 0.0025%, 0.004%, and 0.005% *w*/*v*). These six and a further four concentrations were applied to VK7 2014 (0.01%, 0.02%, 0.04%, and 0.1% *w*/*v*). For the *Anopheles funestus* strains, the range was reduced to three concentrations in addition to the positive and negative controls: 0.0004%, 0.004%, and 0.02% *w*/*v*. Three replicates were performed, each using different generations of each strain such that 60 Kisumu individuals were treated with each concentration of insecticide and 50 for each control. Similarly, at least three replicates totaling 60 VK7 2014 individuals were tested at each concentration. However, only 20 VK7 2014 individuals were tested at 0.1% as a part of range finding where mortality had already reached 100% in lower concentrations. For both Fang and FUMOZ-R strains, at least 90 mosquitoes were tested at each concentration over three replicates. Data sets from 24 h post exposure were used to generate values for lethal doses (LD). All raw bioassay data is available in Appendix A.

### 2.4. Establishing Dose Response Curves

A dose response dataset was established for dinotefuran applied by topical application, as well as by sugar feeding assay in a susceptible strain of *Anopheles gambiae* (Kisumu, [9]) by applying a range of concentrations which gave mortality ranging from 0 to 100%. Topical application gives doses in nanograms per mosquito, converted to nanograms per milligram of mosquito by taking the averages of sample weights of 20 mosquitoes. For the ingestion assay, doses in nanograms per milligram of mosquito were found by estimating the average meal sizes of 10% sugar solution and 0.8% Uranine using fluorimetry (see Appendix B—Quantifying the Average Size of a Sugar Meal Using Fluorescein Sodium Salt (Uranine)). Dose was then inferred through the estimated average meal size of 0.4 μL per feed against the average mosquito mass of 20 individuals per sample.

### 2.5. Calculating LD Values and Resistance Ratios

LD_50_ and LD_95_ values with associated 95% confidence intervals were obtained for each strain by fitting the data to a dose response model (‘drc’ package [11] in R Studio [12].

Susceptibility to dinotefuran was compared between strains by calculating a resistance ratio by dividing the LD_50_ of the pyrethroid resistant strain in each species pair by the LD_50_ of the susceptible strain.

## 3. Results

### 3.1. Establishing Dose Response Curves by Ingestion Assay

An ingestion assay was used to plot dose response curves for orally ingested dinotefuran in a sugar solution. The ratio of LD_50_ values in each pair of strains, Kisumu vs. VK7 2014 and Fang vs. FUMOZ-R, were similar, so only Kisumu and VK7 2014 were selected to establish further dose response curves by the ingestion assay for comparison between the two methods.

The feeding rate between tests ranged from 70 to 98% and 80 to 97% in Kisumu and VK7 2014 cohorts, respectively, and there was no visible trend with dinotefuran concentration (see Appendix A). The LD_50_ for Kisumu was 0.08 (0.06–0.11) ng of dinotefuran per mg of mosquito and the value for VK7 2014 was 0.17 (0.12–0.23) ng of dinotefuran per mg of mosquito (Figure 3), resulting in a resistance ratio of 2.13. Lethal doses (LD_50_ and LD_95_) by ingestion are shown in Table 1.

### 3.2. Establishing Dose Response Curves by Topical Application

Because of the practical challenges in performing ingestion assays, particularly in field settings and at high throughput, dose response curves were also plotted using topically applied dinotefuran in two pairs of *Anopheles* strains as a comparator to the ingestion assay. Topical bioassays (Figure 4A,B) for Kisumu generated an LD_50_ value of 0.75 (0.55–1.03) ng of dinotefuran per mg of mosquito and VK7 2014 assays generated an LD_50_ value of 5.34 (3.97–7.19) ng of dinotefuran per mg of mosquito; together this gives a resistance ratio of 7.12. LD_50_ values for Fang and FUMOZ-R were 2.31 (1.63–3.27) and 7.47 (5.98–9.32) ng of dinotefuran per mg of mosquito, respectively, resulting in a resistance ratio of 3.23. Lethal doses (LD_50_ and LD_95_) by topical application are shown in Table 2.

## 4. Discussion

There is a growing array of vector control tools based on insecticides which act via a range of different exposure routes. The Attractive Targeted Sugar Bait (ATSB) is the only one that involves ingestion by adult mosquitoes. One ATSB currently under evaluation includes dinotefuran, which mosquitoes feed on in a sugar-based bait. As with any insecticide-based intervention there is a need to monitor for the emergence of resistance in the target population, which conventionally has been done using testing methods which expose field caught mosquitoes of the target population via tarsal contact to a treated bottle [5] or filter paper [4]. Because the exposure route of an ingested insecticide is different to a contact insecticide, the results of these tests may not be an accurate indicator of resistance and risk of failure of ATSBs.

This study considered two alternative methods to screen for resistance. The first was an oral ingestion assay developed to prevent tarsal contact while allowing ingestion of a spiked sugar meal, the most direct test for resistance to an oral insecticide. The assay was demonstrated to be robust and quantitative enough to be able to establish a dose response in laboratory strains of *Anopheles*, including in two strains that are highly resistant to pyrethroids. By using a uranine marker, individual mosquitoes that fed were identified, and in the controlled laboratory setting of these experiments the feeding rate was high. However, the applicability of the sugar feeding assay used here in the field is limited due to the large variation of sugar feeding behavior—and thus insecticide uptake—when applied to field caught mosquitoes and using a less controlled laboratory environment. It is likely that the feeding rate, which was high in laboratory strains adapted to feeding on an artificial sugar source, would be much lower in field caught adults or adults emerging from field collected larvae. A low feeding rate would further increase the resources required to produce significant data which could be relied upon in a screen for emerging resistance. Even in these experiments conducted in tightly controlled laboratory conditions and with mosquitoes reared using standardized protocols [9], the results were varied. The methodology would be difficult to standardize sufficiently that it could be performed in multiple field sites, likely with less controlled environments, and achieve robust and comparable data. The ingestion bioassay method also requires greater resources in terms of space and time than topical application, and access to a fluorescent microscope.

The consistently high feeding rate across treatments in this study, which was not correlated with concentration of dinotefuran, suggests a lack of any detectable repellent effect of the dinotefuran. However, in adapting this method for other insecticides, there is a risk of a repellent effect reducing the feeding rate. To avoid this as a possible confounding factor, it is important to use some methods to eliminate individuals that do not feed from mortality scoring, either including uranine and scoring fluorescence as done here or using an alternative such as Trypan blue dye [6,7]. Another possible confounding factor is that it has not been established exactly how long mosquitoes may survive after ingesting particularly lower concentrations of insecticide without direct observation for the whole exposure period. It is possible mosquitoes may have fed just before collection and be scored as survivors when they may have died even minutes later. However, the assay still demonstrated sufficient sensitivity to measure a difference in mortality between concentrations in a dose responsive manner, so this does not appear to prevent the ingestion assay being applicable for this purpose.

Because of the logistical challenges of the ingestion assay, a practical and well established topical application bioassay [14] was also used, applying dinotefuran to the thorax of adult *Anopheles* mosquitoes with organic carrier to bypass lipid cuticle barriers. The direct application of insecticide to the mosquito thorax bypasses the need for uptake of insecticide from a surface and penetration of the insecticide through the cuticle, and mortality as a result of this exposure route has been shown in other insects to correlate well with oral toxicity [8]. The variability of the data between replicates is less with topical application because parameters which define the dose taken up by the mosquito are less variable than for the ingestion assay, producing more robust data.

This study compared the topical application dose response with the respective response of the oral toxicity bioassays, across four *Anopheles* strains. The dose response curves plotted for the same strains were very similar, and there was similar relative susceptibility between the two strains tested with both methods. Topical application is a well-established method and relatively easy to apply [15]. The similarity of results between the two methods demonstrated here suggests that a topical application-based DC, determined based on WHO guidelines, could be used as a proxy for monitoring the development of resistance in field populations to orally ingested dinotefuran from ATSB stations. However, tissue-specific resistance mechanisms are not well studied and there is risk that topical application will not pick up on the emergence of an as yet unidentified ingestion specific mechanism. If potential resistance is observed in results of susceptibility monitoring using topical application, further investigation would be warranted, including exploring such possible mechanisms using ingestion assays in the laboratory. Similarly, the results of topical testing could be affected by the presence of cuticular resistance, through cuticular thickening, altered cuticle composition, or alterations in receptors that affect uptake and penetration of active ingredients. Such resistance mechanisms may be primarily overexpressed in the tissues of the mosquito that are typically in contact with insecticides, such as the tarsi. The direct application of acetone to the thorax (the solvent used to deliver insecticides during topical application) is believed to bypass these mechanisms, and no correlation between cuticular resistance and reduced mortality by topical application has yet been reported. The WHO approach to establishing a DC is to perform dose response experiments and establish LC values for a range of strains susceptible to the insecticide being tested, and then to select the highest DC established for the least susceptible strain, based either on the calculated LC values (DC = 2 × LC_99_) or an observed LC_100_, defined as the lowest concentration tested which reliably produces 100% mortality in susceptible strains [4]. A pragmatic decision may be made as to whether to recommend a specific DC for each species or to select the highest DC to use for a group of species, and sometimes rounding the calculated DC to a value more easily applied in field testing [16]. Based on the dose response observed and LC values calculated in this study, a DC of 100 ng/mosquito would be recommended for topically applied dinotefuran for *An. gambiae* and *An. funestus*. However, a lower tentative DC of incipient resistance, at 10 ng/mosquito, is highly recommended as well, to collect baseline susceptibility data and capture possible variation in bioassay responses among populations in the ATSB trial sites. These trials will further validate the methodology, as well as define the most appropriate DC for screening field *Anopheles* populations.

The relative susceptibility of Kisumu and VK7 2014 strains was measured by each method by calculating a resistance ratio of 5.4 by ingestion and 1.8 by topical application. Both these ratios are very low, all below 10, and do not indicate that there is resistance, but rather inherent variability in susceptibility between strains. A more robust validation of the correlation of results between the two methods by repeating this study with a strain known to be resistant to dinotefuran, or neonicotinoids in general, would help to confirm comparability of results from the two methods but to date no such resistance has been reported in field caught mosquitoes and so no such laboratory strain is available. Another possibility would be to make use of transgenic strains which have resistance to neonicotinoids induced, using a method such as CRISPR/Cas9 [17,18].

The lethal dose of dinotefuran was lower by ingestion than by topical application (approximately 20 times). This points towards a higher toxicity when ingested, though there are several sources of variability in calculating the precise dose of dinotefuran ingested in the sugar feeding assay that mean a direct comparison cannot be made. These include different meal sizes taken by individual mosquitoes, related to body size and previous handling, some individuals taking full sugar meals and some only partial feeds, and the possibility that different volumes are ingested in treatments where insecticide is added.

The volume of bait ingested from an ATSB may be different to the volume of sugar water ingested in this assay, and the size of sugar meal may differ between mosquito populations. It is not, therefore, possible to directly compare the toxic ingested dose of dinotefuran in this assay with the actual dose of dinotefuran in the ATSBs and predict efficacy against target mosquitoes. We can, however, make some estimations based on the assumption that a similar volume of bait is ingested from an ATSB station by wild mosquitoes. The Westham ATSB stations currently under evaluation contain 0.1% dinotefuran, so a mosquito taking up 0.4 μL of bait (Appendix B) will ingest 400 ng of insecticide, 300 times the dose shown to kill 100% of mosquitoes in the ingestion assay. This means that if only 0.0013 μL of bait is ingested, it will be lethal to the mosquitoes. Based on the calculated LC_50_, an amount consumed 3000 times lower than the typical sugar meal would be sufficient to kill 50% of the mosquitoes which feed on it. The bait stations should continue to be effective in populations even where resistance is seen to have emerged through monitoring, using the relatively sensitive DC which has been established.

Practically, no cross-resistance between dinotefuran and pyrethroids was observed. The pyrethroid resistance ratio for these same strains tested with permethrin previously were 145.77 (149–397) via topical application and 128.23 (81.4–198.5) via a tarsal assay [9], but the response of the same strains to dinotefuran was essentially not different or indicated a very low cross-resistance. These data confirm the utility of chemicals belonging to different Insecticide Resistance Action Committee (IRAC) mode of action (MoA) classes [19] (i.e., different target sites and/or routes of uptake for insecticide resistance management (IRM)). Neonicotinoids are nicotinic acetylcholine receptor competitive modulators (IRAC class 4A), with a high selectivity in binding to insect nicotinic acetylcholine receptor sites over that of mammal receptors [20]. The target of action is thus different to pyrethroids, which are sodium channel modulators (IRAC class 3A). None of the WHO Prequalified Vector Control Products contain dinotefuran, and so it is unlikely that mosquitoes have been exposed and developed resistance to dinotefuran. Fludora^®^ Fusion and SumiShield 50WG IRS formulations contain clothianidin, also a neonicotinoid, and their potential for use against pyrethroid resistant insects has been demonstrated [21,22]. Extensive experiments with a proposed diagnostic concentration of 2% *w*/*v* clothianidin on filter papers failed to find conclusive evidence of resistance in 43 sites in sub-Saharan Africa [23], or in *Anopheles arabiensis* in Ethiopia [24]. No evidence of resistance to clothianidin was found in western Kenya using a DC of 150 µg/bottle [25]. However, once an insecticide is being deployed a selection pressure is applied and there is a risk of resistance evolving, and so once validated, the DC should be used to perform regular resistance monitoring in all sites where ATSBs are deployed. Validation of this methodology should also be carried out for any future insecticides used in new ATSB designs. There is also a risk of cross-resistance to dinotefuran as a result of exposure to other neonicotinoids used for vector control or in agriculture.

## 5. Conclusions

An approach has been demonstrated by this study for establishing a suitable method for screening for resistance to a non-contact insecticide.

A discriminating, or diagnostic concentration for topically applied dinotefuran, has been proposed and should now be validated against field mosquito populations where ATSBs are under evaluation. Validation with other ingested insecticides is recommended as further ATSB or similar products are developed.

## Figures and Tables

**Figure 1 insects-13-00311-f001:**
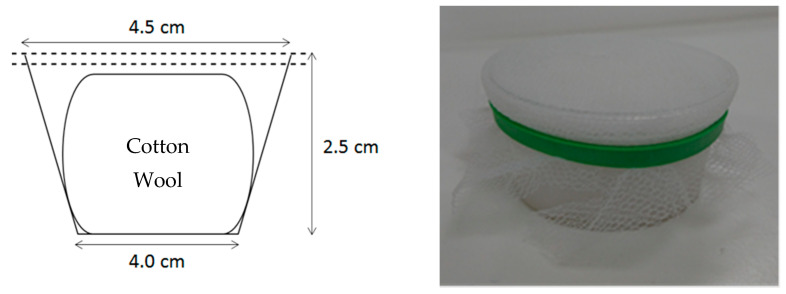
Sugar feeder made from a plastic pot of 2.5 cm height, large radius of 4.5 cm, and small radius of 4 cm. A wad of cotton wool was pressed into the pot to be just below the upper lip without touching the netting (judged by eye) and soaked in sugar solution into which the required concentration of insecticide was dissolved. A double layer of netting was stretched across the top and secured using an elastic band.

**Figure 2 insects-13-00311-f002:**
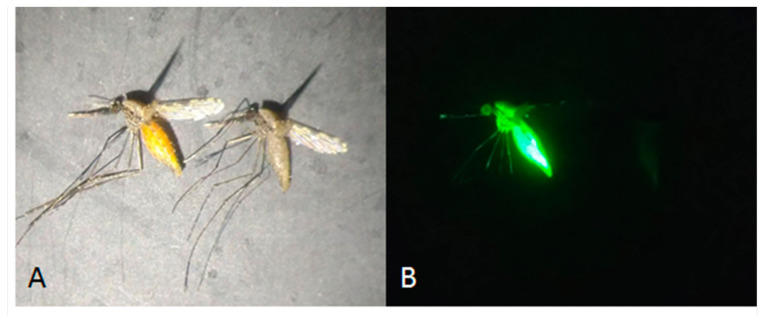
Side by side comparison of two adult female mosquitoes, one fed on 10% sugar solution only (**right**) and one fed on 10% sugar solution with 0.8% Uranine (**left**). Photographs are taken using white light (**A**) and UV light under a YFP filter (**B**).

**Figure 3 insects-13-00311-f003:**
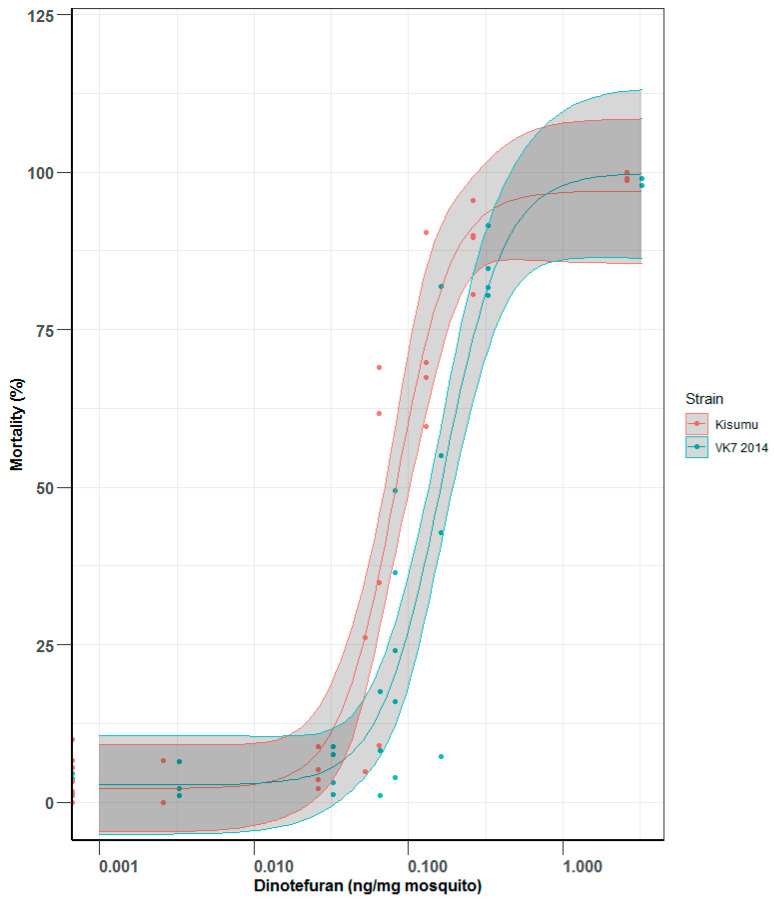
Mortality comparison between Kisumu and VK7 strains when fed on sugar solution spiked with dinotefuran at known concentrations, resulting in doses of dinotefuran in nanograms per milligram of mosquito. Central lines of each curve represent the dose response of each species. Black lines indicate LD_50_ values; red refers to the mortality curve of Kisumu; blue refers to the mortality curve of VK7 2014. The shaded areas of each curve represent 95% CI values, generated by R software using the ggplot2 package [13].

**Figure 4 insects-13-00311-f004:**
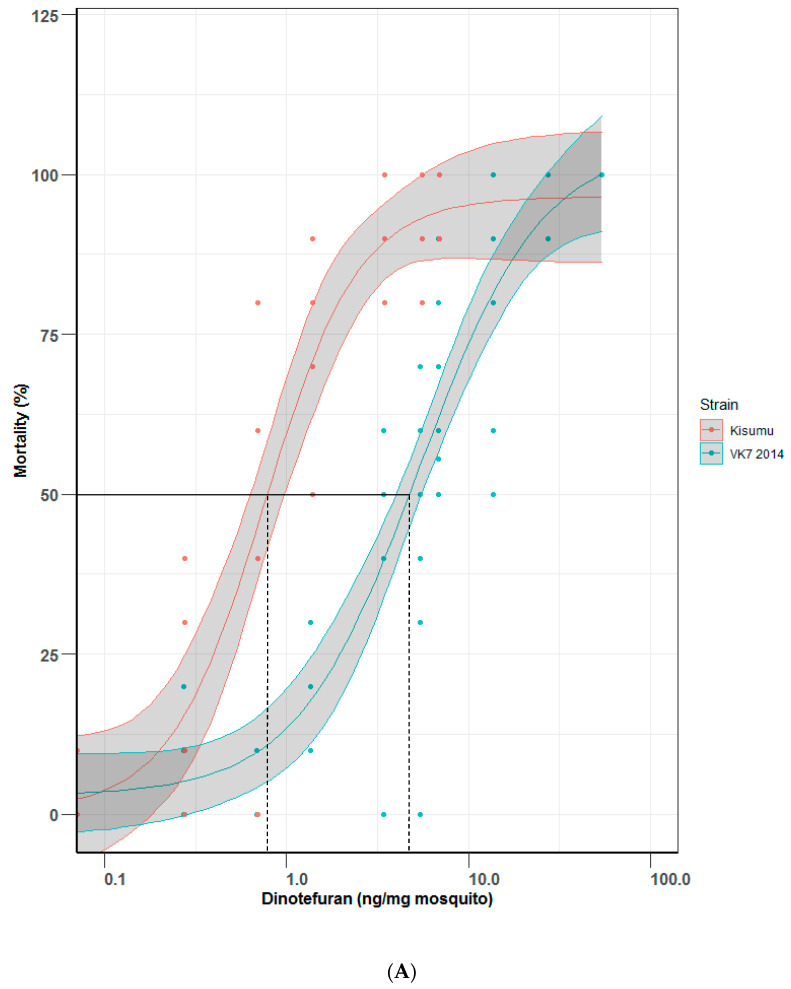
Mortality comparisons between (**A**) Kisumu and VK7 2014 strains and (**B**) Fang and FUMOZ-R strains in a topical application bioassay. Central lines of each curve represent the dose response of each species. Black lines indicate LD_50_ values on both graphs; red lines show mortality curves of the insecticide susceptible strains of each pair of strains (Kisumu and Fang); blue lines show data for insecticide-resistant strains (VK7 2014 and FUMOZ-R). Shaded areas of each curve represent 95% CI values, generated by R software using the ggplot2 package [13]. (**A**) omits data for VK7 2014 treated with 136 ng per mg of mosquito, this gave 100% mortality with no variance as did the highest represented range (54 ng per mg of mosquito, refer to Appendix A) and so was removed for clarity.

**Table 1 insects-13-00311-t001:** Lethal doses and lethal concentrations of dinotefuran ingested in a sugar solution in two strains of *Anopheles* mosquitoes. 95% CI given in parentheses. Kisumu is a lab strain of *Anopheles gambiae*, VK7 2014 is *An. coluzzii*.

Strain	LD_50_	LC_50_	LD_95_	LC_95_
	ng/mg of Mosquito	ng per Mosquito	ng/mg of Mosquito	ng per Mosquito
Kisumu	0.08 (0.06–0.11)	0.12 (0.09–0.17)	0.29 (0.12–0.67)	0.45 (0.19–1.04)
VK7 2014	0.17 (0.12–0.23)	0.2 (0.15–0.28)	0.65 (0.3–1.38)	0.79 (0.37–1.69)

**Table 2 insects-13-00311-t002:** Lethal doses and lethal concentrations in four strains of *Anopheles* mosquitoes by topical application of dinotefuran. 95% CI given in parentheses. Kisumu is a lab strain of *Anopheles gambiae*, VK7 2014 is *An. coluzzii*, and Fang and FUMOZ-R are *An. funestus*.

Strain	LD_50_	LC_50_	LD_95_	LC_95_
	ng/mg of Mosquito	ng per Mosquito	ng/mg of Mosquito	ng per Mosquito
Kisumu	0.75 (0.55–1.03)	1.09 (0.80–1.49)	4.41 (1.78–10.93)	6.38 (2.57–15.82)
VK7 2014	5.34 (3.97–7.19)	7.85 (5.84–10.57)	52.35 (18.79–145.86)	76.96 (27.62–214.41)
Fang	2.31 (1.63–3.27)	2.43 (1.72–3.43)	19.64 (9.28–41.57)	20.62 (9.74–43.65)
FUMOZ-R	7.47 (5.98–9.32)	6.72 (5.38–8.39)	31.82 (3.69–274.05)	28.64 (3.33–246.65)

## Data Availability

All raw data can be found in the Appendix A.

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
