# Peer review of "A Practical Insecticide Resistance Monitoring Bioassay for Orally Ingested Dinotefuran in Anopheles Malaria Vectors"

_insects, 2022, doi:10.3390/insects13040311_

Round 1

Reviewer 1 Report

Overall, this paper seems to have a lot of potential and the topic they investigate by comparing between ingestion and topical application assays is quite interesting and could be very useful when assessing resistance to ATSBs. However, I am not entirely convinced the authors have supported their claim that there is a similarity between the resistance ratios of the ingestion vs topical assays, and that the topical assay could be used as a valid substitute for the ingestion assays to accurately measure susceptibility/resistance to ATSBs. There are multiple areas that require revision or need to be addressed prior to publication.

One of the major concerns I have with this article is that they had complications with obtaining confidence intervals for some of their calculations. I am not sure if this can be remedied somehow, either by using a different software program or by the authors trying to understand why the data is messy enough to cause this problem. However, it is critical to have these confidence intervals especially for some of the calculations regarding the resistance ratios in order to determine if there is a statistically significant difference or not between the levels of resistance determined between the two assay types.

The authors ultimately make the claim that topical susceptibility assays with dinotefuran are better than sugar ingestion assays, for the myriad of reasons about the difficulties and variances of the latter connected with sugar feeding in the lab and field. However, they then jump to the other conclusion that the two assays are similar anyways, but the data do not seem to support this claim.  The topical assay for Kisumu gives an LC50 of 0.83 while the ingestion assay LC50 is nearly 10X less at 0.08.  Similar differences were seen with VK7, and the RRs between the strains were 2 for the ingestion vs. 5.4 for the topical.  This is more than a 2X difference. I cannot see how this suggests topical assays are equivalent or reflective of sugar ingestion assays for a ATSB active ingredient.  Furthermore, dinotefuran will not always be the AI in ATSBs, and it is not a good assumption that the digestion process would not affect the results relative to topical application. The danger here is that topical applications become the primary method to measure susceptibility for ATSBs, but then what if a weird or unusual crop- or midgut-based resistance mechanisms arises, and it gets completely missed for months/years while investigators are only doing topical based assays.  

Another concern is that the Kisumu and the VK7 2014 strains are different species but are compared directly as if they are the same species of Anopheles. Please acknowledge this and comment if this has an impact on the results. I’m curious to know why the authors did not choose to compare the same species here, did they just happen to be the strains/species of mosquitoes they had access to? Possibly similarly, do the authors know that FANG and FUMOZ are the same species of An. funestus group?...I know that FANG derives from Angola and FUMOZ from Mozambique, but not sure on this, and I don’t see in the text where the authors described nature of the resistance of FUMOZ-R (e.g. pyrethroids or other?).

The authors also do not report a “discriminating dose” in the tables as they said they did, but rather determined this in the Discussion. However, it not clear that they actually determine this discriminating dose appropriately according to the WHO guidelines because it is clear they used the “pragmatic”
 method they refer to in the Discussion, but I’m not familiar with this in the WHO literature.

General fixes for the paper:

A lot of investigators refer to ATSBs as “Attractive Toxic Sugar Bait”, not “Attractive Targeted Sugar Bait.” I assume this is Westham’s registered trade name, but toxic sugar baits might go beyond Westham’s product, so the author should explain the difference.

Please correct/distinguish LCxx and LDxx in your text and figures.  You are reporting dose by reporting ng/mg of mosquito – so this should be a LD50 or LD90, but your also reporting ng per mosquito, which is a concentration, and so should be reported as a LC50 or LC90.

Fix and check your units for your reported concentrations/doses. There are multiple areas where you report them as just “ng” when it should be “ng/mg of mosquito” or “ng per mosquito”

Other grammar edits and minor revisions (this would have been easier to review is line number were incorporated):

Abstract

Define what DD is.

Introduction

(Paragraph 2)

Define what “IVCC” means

(Paragraph 3)

Fix first sentence, rewrite as “Dinotefuran is the active ingredient in these ATSBs” for it to be more clear and concise.

(Paragraph 4)

Define what negative log P is to the reader.

First sentence, remove “and”

(Paragraph 5)

Define “AI”

Materials and methods

(Section 2.1)

Define “LITE”

Second sentence, change the word “multiply” to “multiple”

(Section 2.2)

Were there multiple replicates completed for the ingestion assay? Please clarify how many.

So the feeding cups were inside the cages for 24 hrs and the mosquitoes allowed to feed for that entire time, and then after that 24 hour “exposure” period, the mosquitoes were immediately aspirated out and deemed as alive or dead? If yes, I have concerns that the mosquitoes fed at different time points during the 24 hour exposure period. For example, some mosquitoes could have possibly fed immediately before the feeding period ended, while others fed immediately after the cups were put in the cages, and therefore were not given the same amount of time to process/digest the insecticide. Please clarify and address this as a possible area of weakness when comparing the two assays, since the mosquitoes for the topical application were all assessed 24 hrs after exposure, while the ingestion assayed mosquitoes could’ve varied with the time point at which they were collected and assessed after feeding.

Figure 1, consider labeling the soaked Cotton wool In the diagram, for clarity

(Section 2.3)

How many mosquitoes were tested for each control group in the VK 7 and the Anopheles funestus strains? Please clarify.

Please also clarify the number of replicates completed for each of the strains tested. It seems like the words “samples” and “cohorts” were used in this paragraph in a confusing manner and should actually be replaced by the word “replicates”

(Section 2.5)

Perhaps use the term “resistance ratio” (RR) in place of “ratio of susceptibility.” This term is more commonly used in the literature and more accurately describes what you are calculating.

Results

(Section 3.1)

Second paragraph, “Table 2” should actually be “Table 1” based on the info you are referring to.

Tables 1 and 2, please report the LC50s instead of (or in addition to) the LC95s, especially since these values are easier to see in the figures. Italicize “Anopheles”

Figures 3 and 4, add black lines to graph to show where LC50s are – I’m assuming these black lines are vertical interpolation lines that you meant to add. Is the center line for each strain the actual regression curve? Define the shaded area  - I assume it represents the confidence interval? But you state that the confidence intervals couldn’t be calculated… please edit the graph or clarify them better in the figure legends.

(Section 3.2)

Also calculate the resistance ratios confidence intervals (divide the CIs of the resistant strain by the CIs of the susceptible strain). You should also do this for the topical assays. These ratios between the two assay types and whether or not their CIs overlap is critical to determine if the assays can actually produce similar/comparable results.

Discussion

Throughout discussion, hyphenate “cross-resistance”

Second paragraph, “a low response rate” should be changed to “a low feeding rate”

Fifth paragraph, Italicize “Anopheles”

Sixth paragraph, first sentence, I think “tolerance” should be replaced with “susceptibility.” Need confidence intervals of  resistance ratios to see if there is a significant difference between the ratios calculated between the two assay types.

Seventh paragraph, “individuals” is misspelled

Maybe discuss or mention the difference between the mode of action between pyrethroid vs neonicitinoid insecticides in order to better explain why cross-resistance between these types is unlikely. What about other insecticides that might have similar modes of action to dinotefuran? Is there a concern for cross resistance with those?

Also, you may want to explain that tarsal contact when mosquitoes are feeding on ATSBs in the field is prevented (I think?), because of the design of the Westham ATSB stations. Correct?

Another thing to mention/discuss is how might cuticle resistance impact your results. This type of resistance is not studied quite as much, but I believe it should be addressed here. For example, if you happen to be testing strains that (unknowingly) have different levels of cuticle resistance, this would impact your results if you were to use the topical assay rather than the ingestion assay.

Author Response

Overall, this paper seems to have a lot of potential and the topic they investigate by comparing between ingestion and topical application assays is quite interesting and could be very useful when assessing resistance to ATSBs. However, I am not entirely convinced the authors have supported their claim that there is a similarity between the resistance ratios of the ingestion vs topical assays, and that the topical assay could be used as a valid substitute for the ingestion assays to accurately measure susceptibility/resistance to ATSBs. There are multiple areas that require revision or need to be addressed prior to publication.

  • We acknowledge that there are limitations to the study and that further validation is needed for this method, which, as we describe in the manuscript, is ongoing as the proposed discriminating doses and methodology are used in the field.

One of the major concerns I have with this article is that they had complications with obtaining confidence intervals for some of their calculations. I am not sure if this can be remedied somehow, either by using a different software program or by the authors trying to understand why the data is messy enough to cause this problem. However, it is critical to have these confidence intervals especially for some of the calculations regarding the resistance ratios in order to determine if there is a statistically significant difference or not between the levels of resistance determined between the two assay types.

  • Due to the variability in the data from the ingestion assay we were unable to calculate CI with the original Poloplus software Following the reviewer’s suggestion we switched to using the ‘drc’ package in R software to calculate LDs and 95% CI values The resulting values do not differ substantively from those calculated in Poloplus but were able to calculate CI values where we were unable to before.
  • We do not claim that there is no statistically significant difference between data from the two methods. As discussed below, the resistance ratios are so small that they are not biologically significant and so it is not important to compare them statistically.

The authors ultimately make the claim that topical susceptibility assays with dinotefuran are better than sugar ingestion assays, for the myriad of reasons about the difficulties and variances of the latter connected with sugar feeding in the lab and field. However, they then jump to the other conclusion that the two assays are similar anyways, but the data do not seem to support this claim.  The topical assay for Kisumu gives an LC50 of 0.83 while the ingestion assay LC50 is nearly 10X less at 0.08.  Similar differences were seen with VK7, and the RRs between the strains were 2 for the ingestion vs. 5.4 for the topical.  This is more than a 2X difference. I cannot see how this suggests topical assays are equivalent or reflective of sugar ingestion assays for a ATSB active ingredient. 

  • The ‘RR’ that we report are negligible; a difference between 0.8 and 0.08 might be 10x but is not biologically significant, particularly given the noisiness of the data. A challenge in determining that the methods are equivalent is that we don’t have a neonicotinoid-resistant strain to confirm the similarity of RR between methods, as we discuss in the manuscript.

Furthermore, dinotefuran will not always be the AI in ATSBs, and it is not a good assumption that the digestion process would not affect the results relative to topical application. The danger here is that topical applications become the primary method to measure susceptibility for ATSBs, but then what if a weird or unusual crop- or midgut-based resistance mechanisms arises, and it gets completely missed for months/years while investigators are only doing topical based assays.  

  • The reviewer is correct that there may be a novel or as yet unidentified mechanism of resistance to ingested insecticide that a topical application screen would fail to identify. If potential resistance is observed in results of susceptibility screening in the future further investigation would be warranted including exploring possible mechanisms and using ingestion assays in the lab. A sentence has been added to the Discussion to mention this. But previous work has suggested that resistance to the two routes of entry correlate well in other species, and this study aimed to demonstrate this in mosquitoes. We also added mention in the Conclusion that the proposed methodology be validated for any future insecticides used in ATSBs.

Another concern is that the Kisumu and the VK7 2014 strains are different species but are compared directly as if they are the same species of Anopheles. Please acknowledge this and comment if this has an impact on the results. I’m curious to know why the authors did not choose to compare the same species here, did they just happen to be the strains/species of mosquitoes they had access to?

  • Kisumu is included in many studies at LSTM as a model lab colony originating from the Anopheles gambiae species complex which is susceptible to all classes of insecticides. We do not hold a similar colony which would allow a direct species comparison with VK7 2014. The wording has been adjusted in the Methods and Results accordingly, and more detail about each of the colonies has been added, with a reference to Williams et al 2020.

Possibly similarly, do the authors know that FANG and FUMOZ are the same species of An. funestus group?...I know that FANG derives from Angola and FUMOZ from Mozambique, but not sure on this, and I don’t see in the text where the authors described nature of the resistance of FUMOZ-R (e.g. pyrethroids or other?).

  • FUMOZ-R was generated as a resistant strain by selection of the Fang strain, and so they are a much closer pair of strains for comparison. More detail about each of the colonies has been added to the Methods, with a reference to Williams et al 2020.

The authors also do not report a “discriminating dose” in the tables as they said they did, but rather determined this in the Discussion. However, it not clear that they actually determine this discriminating dose appropriately according to the WHO guidelines because it is clear they used the “pragmatic” method they refer to in the Discussion, but I’m not familiar with this in the WHO literature.

  • Thank you for identifying this error, which is a leftover from a previous draft of the manuscript. Reference to DDs being in the tables have been removed. The ‘pragmatic approach’ is in line with WHO methodologies for establishing discriminating doses. The report of the study to establish DDs for mosquitoes is currently in preparation and a citation has been added to reflect this – ref [19].

General fixes for the paper:

A lot of investigators refer to ATSBs as “Attractive Toxic Sugar Bait”, not “Attractive Targeted Sugar Bait.” I assume this is Westham’s registered trade name, but toxic sugar baits might go beyond Westham’s product, so the author should explain the difference.

  • This difference in terminology has been clarified in the Introduction with relevant citations.

Please correct/distinguish LCxx and LDxx in your text and figures.  You are reporting dose by reporting ng/mg of mosquito – so this should be a LD50 or LD90, but your also reporting ng per mosquito, which is a concentration, and so should be reported as a LC50 or LC90.

  • In checking again with related literature, we have elected to stick with reporting doses (ng/mg of mosquito, as suggested by the reviewer) instead of concentrations, though have made sure concentration data is still available.

Fix and check your units for your reported concentrations/doses. There are multiple areas where you report them as just “ng” when it should be “ng/mg of mosquito” or “ng per mosquito”

  • Units have been clarified throughout the manuscript.

  • All other grammar edits and minor revisions have been addressed.

(Section 2.2)

Were there multiple replicates completed for the ingestion assay? Please clarify how many.

  • This has now been described more clearly in the Methods.

So the feeding cups were inside the cages for 24 hrs and the mosquitoes allowed to feed for that entire time, and then after that 24 hour “exposure” period, the mosquitoes were immediately aspirated out and deemed as alive or dead? If yes, I have concerns that the mosquitoes fed at different time points during the 24 hour exposure period. For example, some mosquitoes could have possibly fed immediately before the feeding period ended, while others fed immediately after the cups were put in the cages, and therefore were not given the same amount of time to process/digest the insecticide. Please clarify and address this as a possible area of weakness when comparing the two assays, since the mosquitoes for the topical application were all assessed 24 hrs after exposure, while the ingestion assayed mosquitoes could’ve varied with the time point at which they were collected and assessed after feeding.

  • This point is valid, and has been discussed in the Discussion. However, we have shown that the assay is sufficiently sensitive as it stands to measure a difference in mortality between concentrations in a dose responsive manner, and the feeding rate is consistently high, so it is not a weakness of the assay for this purpose. More experiments could be done to further understand the biology of the insecticide, but would not be relevant to the question at hand.

Figure 1, consider labeling the soaked Cotton wool In the diagram, for clarity

  • Done

(Section 2.3)

How many mosquitoes were tested for each control group in the VK 7 and the Anopheles funestus strains? Please clarify.

  • This is now clarified in the Methods section.

Please also clarify the number of replicates completed for each of the strains tested. It seems like the words “samples” and “cohorts” were used in this paragraph in a confusing manner and should actually be replaced by the word “replicates”

  • This is now clarified in the Methods.

(Section 2.5)

Perhaps use the term “resistance ratio” (RR) in place of “ratio of susceptibility.” This term is more commonly used in the literature and more accurately describes what you are calculating.

  • Resistance ratio is indeed the more accepted terminology. Since the RR was negligible in this case, and to make it clear that we have not identified any resistance in these strains we had used the phrase ‘ratio of susceptibility’. On the suggestion of the reviewer we have reverted to ‘resistance ratio’.

Results

(Section 3.1)

Second paragraph, “Table 2” should actually be “Table 1” based on the info you are referring to.

  • Numbering checked throughout and corrected where needed.

Tables 1 and 2, please report the LC50s instead of (or in addition to) the LC95s, especially since these values are easier to see in the figures. Italicize “Anopheles”

  • LC50s are apparent in the figures, but have now been added to the Tables for clarity. Italicisation checked throughout and corrected where needed.

Figures 3 and 4, add black lines to graph to show where LC50s are – I’m assuming these black lines are vertical interpolation lines that you meant to add. Is the center line for each strain the actual regression curve? Define the shaded area  - I assume it represents the confidence interval? But you state that the confidence intervals couldn’t be calculated… please edit the graph or clarify them better in the figure legends.

  • We have added black lines onto the graphs and referred to the centre lines as the dose response curve. We have also defined the shaded area as the 95% CI, as the reviewer pointed out.

(Section 3.2)

Also calculate the resistance ratios confidence intervals (divide the CIs of the resistant strain by the CIs of the susceptible strain). You should also do this for the topical assays. These ratios between the two assay types and whether or not their CIs overlap is critical to determine if the assays can actually produce similar/comparable results.

  • As discussed above, the data is variable and we were initially unable to calculate CIs in some cases using the original Poloplus software, but were able to using R Studio. All CIs are now presented. However, the resistance ratios are not biologically significant, as discussed above, and so we do not believe that further statistical analysis is warranted. We tried the calculation suggested by the reviewer, but do not believe that the results are useful for comparing results and complicate description of the results and so have not included them.

Discussion

Throughout discussion, hyphenate “cross-resistance”

  • Done

Second paragraph, “a low response rate” should be changed to “a low feeding rate”

  • Done

Fifth paragraph, Italicize “Anopheles”

  • Done

Sixth paragraph, first sentence, I think “tolerance” should be replaced with “susceptibility.”

  • Done

Seventh paragraph, “individuals” is misspelled

  • Done

Maybe discuss or mention the difference between the mode of action between pyrethroid vs neonicitinoid insecticides in order to better explain why cross-resistance between these types is unlikely. What about other insecticides that might have similar modes of action to dinotefuran? Is there a concern for cross resistance with those?

  • We have added a sentence to the Discussion to describe the mode of action of neonicotinoids and highlight the difference to the MoA of pyrethroids. The potential for cross-resistance between dinotefuran and other neonicotinoids is already addressed at the end of the Discussion.

Also, you may want to explain that tarsal contact when mosquitoes are feeding on ATSBs in the field is prevented (I think?), because of the design of the Westham ATSB stations. Correct?

  • We have added a sentence to the Discussion to say that although tarsal contact is possible while mosquitoes feed on an ATSB, the Westham ATSB stations are designed to minimise this.

Another thing to mention/discuss is how might cuticle resistance impact your results. This type of resistance is not studied quite as much, but I believe it should be addressed here. For example, if you happen to be testing strains that (unknowingly) have different levels of cuticle resistance, this would impact your results if you were to use the topical assay rather than the ingestion assay.

  • This is a good point, and a sentence has been added to the Discussion to address it. In fact we believe that cuticular resistance mechanisms (now referred to and cited in the Discussion) are focused mainly on the areas of the mosquito which contact insecticide, the tarsi, involving cuticular thickening or altered cuticular composition, as well as altered receptors affecting uptake and penetration. We believe that direct application of insecticide in acetone to the thorax bypasses these mechanisms. It is possible that cuticular resistance affects the results of topical application, but we have not seen evidence of it in our work with multiple resistant strains.

Reviewer 2 Report

This work is important for moving the idea of targeting the sugar-feeding behavior of mosquitoes for control purposes.  Developing methods and protocols for evaluating resistance to control methods through mosquito ingestion will allow for mosquito control programs to directly asses the utility of the ATSBs.  

ATSB methods finally have a product that is being evaluated in Africa using a new mode of action for mosquito control.  One reason this method is desirable is because it has been claimed to mitigate insecticide resistance. This paper states that because the pesticide is being ingested and not topically delivered as is most mosquito control products how do we monitor for IR?  This methodology laid out by the authors is important and can be used if other active ingredients are being used in ATSBs in the future.  It is not ground breaking science but certainly important for those in the field and for programs that do IR testing.  

As mentioned above it provides a protocol for public health professionals to follow to monitor IR in areas where these ATSBs are being used. This is an important validation. 

The conclusions are consistent with the evidence and arguments presented and they address the main question posed. 

The references are appropriate

Author Response

We thank the reviewer for recognising the value of this study and for their positive comments.